# Validation of a Dendritic Cell and CD4+ T Cell Restimulation Assay Contributing to the Immunogenicity Risk Evaluation of Biotherapeutics

**DOI:** 10.3390/pharmaceutics14122672

**Published:** 2022-12-01

**Authors:** Michel Siegel, Guido Steiner, Linnea C. Franssen, Francesca Carratu, James Herron, Katharina Hartman, Cary M. Looney, Axel Ducret, Katharine Bray-French, Olivier Rohr, Timothy P. Hickling, Noel Smith, Céline Marban-Doran

**Affiliations:** 1Roche Pharmaceutical Research and Early Development, Pharmaceutical Sciences, Roche Innovation Center Basel, 4070 Basel, Switzerland; 2Lonza Biologics, Chesterford Research Park, Saffron Walden CB10 1XL, UK; 3UR 7292, IUT Louis Pasteur, Université de Strasbourg, 67300 Schiltigheim, France

**Keywords:** immunogenicity, immunomodulation, biotherapeutics, in vitro T cell assay, assay validation

## Abstract

Immunogenicity, defined as the ability to provoke an immune response, can be either wanted (i.e., vaccines) or unwanted. The latter refers to an immune response to protein or peptide therapeutics, characterized by the production of anti-drug antibodies, which may affect the efficacy and/or the safety profiles of these drugs. Consequently, evaluation of the risk of immunogenicity early in the development of biotherapeutics is of critical importance for defining their efficacy and safety profiles. Here, we describe and validate a fit-for-purpose FluoroSpot-based in vitro assay for the evaluation of drug-specific T cell responses. A panel of 24 biotherapeutics with a wide range of clinical anti-drug antibody response rates were tested in this assay. We demonstrated that using suitable cutoffs and donor cohort sizes, this assay could identify most of the compounds with high clinical immunogenicity rates (71% and 78% for sensitivity and specificity, respectively) while we characterized the main sources of assay variability. Overall, these data indicate that the dendritic cell and CD4+ T cell restimulation assay published herein could be a valuable tool to assess the risk of drug-specific T cell responses and contribute to the selection of clinical candidates in early development.

## 1. Introduction

Despite success in the clinic, a substantial number of biotherapeutics elicit unwanted immune or immunogenic responses—termed immunogenicity. One of the hallmarks of immunogenicity is the onset of anti-drug antibodies (ADAs). Due to ADAs exhibiting major consequences for both patient’s safety and treatment efficacy, it is of utmost importance to assess this risk as early as possible during drug development [1,2].

Partially or fully humanized biotherapeutics (i.e., antibodies with minimal non-germline amino acid sequences) are usually at lesser risk of an unwanted immunogenicity response; however, this measure may not completely abrogate ADA formation. It is now established that a compound immunogenicity risk assessment must include multiple complex factors ranging from product-related risks, such as protein structure, formulation, or impurities [3]; patient and disease-related factors, including genetic factors, age, concomitant treatment; and route of administration [4]. In the case of immunomodulatory drugs, adverse events may also be caused by target binding in healthy tissues, or enhanced pharmacology attenuating the activity of target molecules on cells.

Consequently, an integrated preclinical risk assessment should be considered as a key element in biotherapeutics development. Regulatory bodies, such as the European Medicines Agency (EMA) and the Food and Drug Administration (FDA), are now encouraging drug developers to consider risk factors related to the product and to the patient, mentioned above, as early as possible in the development process. An integrated approach relies on the use of specific tools and methods to identify relevant immunogenicity factors and to develop corresponding risk mitigation strategies [5]. Currently, these tools include in silico screening algorithms to scan for sequence liabilities, in vitro cell-based assays to measure various readouts from the immune response (dendritic cell internalization, activation and presentation, T cell activation), and the use of transgenic animal models designed to study the intimate mechanisms of an immune response from a mechanistic viewpoint [6]. However, most of these tools have not undergone a formal qualification process, and factors contributing to assay variability are not always understood. For example, T cell-dependent responses are the major drivers for immunogenicity, and in vitro T cell assays are frequently used to identify and measure CD4+ T cell responses to biotherapeutics. These assays have been derived in different formats and reviewed elsewhere [6,7,8]. However, the sensitivity of these assays is usually quite low, as the size of the pre-existing CD4+ T cell repertoire reactive to the drug is very small, ranging from 1 to 10 cells out of 10^8^ T cells [9].

Here, we describe and characterize a dendritic cell and CD4+ T cell restimulation assay and discuss the potential of such an assay to assess a CD4+ T cell-driven immunogenicity risk. This assay consists of a co-culture between monocyte-derived dendritic cells (moDCs) and autologous CD4+ T cells, including a re-stimulation step to increase assay sensitivity. The main goals of this study were to establish an assay threshold to distinguish between positive and negative responses, to determine the optimal cohort size for the assay, and to identify factors affecting assay variability. We are currently using this assay as part of an integrated approach to rank candidate biotherapeutics during the initial selection process, enabling the selection of lower-risk clinical leads for subsequent large-scale production and clinical trials.

## 2. Materials and Methods

### 2.1. Compounds

Stock solutions of keyhole limpet hemocyanin (KLH-Imject Maleimide-Activated mcKLH, Thermo Fisher Scientific, Basel, Switzerland, #77600) were reconstituted and stored at −80 °C in single-use aliquots according to the manufacturer’s recommendations under sterile conditions. All biotherapeutics were bought from Runge Pharma GmbH & Co (Lörrach, Germany) in their respective formulation and stored according to the manufacturer’s recommendations. Peptides were synthesized by Cambridge Research Biochemicals and reconstituted in sterile ultra-pure water (Invitrogen, Basel, Switzerland, #10977015) and 50% Acetonitrile (≥99.95%, VWR, #83639.320). Biotherapeutics were used at a final concentration of 0.3 μM (peptides were used at a final concentration of 10 μg/mL) for both the DC stimulation stage and re-stimulation stage.

### 2.2. Healthy Donor Cohort

Healthy donors were recruited at Phase I clinical trial units in the UK. All samples were collected under an ethical protocol approved by a local Research Ethics Committee (reference number: 21/LO/0474), and written informed consent was obtained from each donor prior to sample donation. All samples were stored according to the terms of Lonza’s Human Tissue Authority license for the use of samples in research. Peripheral blood mononuclear cells (PBMC) from healthy donors were prepared from whole blood or leukopaks using Lymphorep density gradient medium (Cedarlane, # CL5120) within six hours of blood withdrawal. PBMC were controlled-rate frozen and stored in vapor-phase nitrogen at −196 °C until used in the assays. The quality and functionality of each PBMC preparation were analyzed after seven days of activation, with positive controls such as KLH to assess naïve T cell responses. For each screen, the donor cohorts consisted of typically 30 donors selected to represent the world population in terms of their HLA-DRB1 allele frequency distribution [5] (Appendix A).

### 2.3. DC:CD4+ Re-Stimulation Assay (Epibase^®^IV, Lonza)

Monocytes were isolated from frozen PBMC samples by magnetic bead selection using CD14 microbeads (Miltenyi Biotec # 130-050-201 on an AutoMACS Pro system) and differentiated into immature DC (iDC) using 1000 IU/mL of granulocyte-macrophage colony-stimulating factor (GM-CSF) and 1000 IU/mL of IL-4 in a serum-free mediun (CellGenix # 20805-0500, supplemented with 0.05 mg/mL Gentamicin Lonza # 17-518L) for 5 days at 37 °C, 5% CO_2_. iDC were then harvested, washed and loaded with each test protein/peptide individually for 4 h at 37 °C, 5% CO_2_. A DC maturation cocktail containing TNFα (800 IU/mL) and IL-1β (100 IU/mL) was then added for a further 40–42 h to activate/mature the DC (mDC). The expression of key DC surface markers (CD11c-3.9, CD14-63D3, CD40-5C3, CD80-2D10, CD83-HB15E, CD86-BU63, CD209-9E9A8 and HLA-DR-L243) at both the immature and mature stage were assessed by flow cytometry (Bio-Rad ZE5 Cell Analyzer) to ensure the DC were activated prior to T cell interaction. After a thorough washing procedure, 100,000 mDCs were then co-cultured with 1 million autologous CD4+ T cells (isolated by magnetic bead selection, Miltenyi Biotec # 130-045-101 on an AutoMACS Pro system) in a deep-well plate (final volume of 1.2 mL, Greiner # 780271). The DC:CD4+ T cells ratio is 1:10 and the co-culture is incubated for 6 days at 37 °C, 5% CO_2_ in a humidified atmosphere. On day 6, autologous monocytes were isolated from PBMC using magnetic bead selection (Miltenyi Biotec # 130-050-201 on an AutoMACS Pro system) and loaded with the selected protein or peptide that were initially used to load the DC. After incubation at 37 °C, 5% CO_2_ in a humidified atmosphere for 4 h, the monocytes were washed and then added to anti-IFN-γ/anti-IL-5 pre-coated FluoroSpot plates (Mabtech # FSP-0108-10) along with the corresponding DC:CD4 co-culture in quadruplicate (25,000 monocytes: 250,000 CD4+ T cells in a final volume of 200 µL). The FluoroSpot plates were incubated for 40–42 h at 37 °C, 5% CO_2_ in a humidified atmosphere. After incubation, the FluoroSpot plates were developed according to the manufacturer’s procedure (IRIS FluoroSpot reader, Mabtech) and the number of spot-forming cells (SFC) per well were assessed for each test condition in an automated and unbiased manner.

### 2.4. Data Analysis

Data management and statistical analysis were performed in the R programming language (https://www.R-project.org/, accessed on 28 October 2022, versions 3.6.1 up to 4.1.2), including essential packages for handling generalized linear models (nlme, emmeans) and carrying out variance component analyses (VCA, version 1.4.3).

The calculation of Stimulation Indices (SI) was performed as follows. Spot forming cells (SFC) from the FluoroSpot assay were transformed to a log2 scale, and a generalized linear model (GLM) was applied to estimate the SI (i.e., the ratio between a treatment condition and the donor-matched blank on a linear signal scale) and associated confidence intervals. Quadruplicate SFC measurements were implicitly aggregated by the GLM to yield one SI value for each combination of a specific test compound, donor, and screen. The screens were analyzed sequentially and independently from each other, with the linear model considering a specific cytokine readout of an entire screen as input. The processing workflow was tailored to address a few peculiarities of the given data. Specifically, we used an exponential type of heteroscedasticity adjustment in the GLM to achieve scale-invariance of residuals and injected some Gaussian noise at the low end of the SFC scale to support model convergence with the frequent presence of ties of discrete values around zero. (The standard deviation of this normally distributed, zero-centered noise was chosen to correspond to the replicate variability inferred by the GLM in the limit of zero SFC counts at the low end of the SFC scale and drops down exponentially by a factor of exp (−2) = 0.14 for every unit increase of the log2 SFC). Furthermore, we observed a consistent trend in the data to the effect that higher blank values of a donor corresponded to systematically lower SI values for that donor. The relation between ‘pre-stimulation’ of the blank and observed stimulation indices could be well captured by linear regressions performed for each treatment within a screen. We corrected the raw SI values then for every donor-treatment pair with the respective linear model, basically extrapolating to the value which would have been observed with a common blank value of 0.

Standard quality control plots were generated for every data set, including the visualization of DC differentiation markers, the reproducibility of reference compound data across studies, and (if possible) the variability of repeated compound testing with the same donor. We also looked at the individual stimulation profile of each donor within a study, as the overall inducibility of T-cell response could vary from person to person; simultaneously, this enabled us to rule out the presence of generally inert sample material. A donor response was recorded as “positive” if a SI fold-change of 2 or above (compared to its blank control) was measured at a statistical significance of *p* < 0.05 (using non-adjusted *p*-values from the GLM). The fraction of positive donor responses (within a cohort of typically 30 healthy donors per screen) provided the response rate for the treatment in a specific screen.

## 3. Results

### 3.1. DC:CD4+ T Cell Restimulation Assay Workflow

The general workflow of the assay is illustrated on Figure 1a. Test items were investigated in independent screens of the DC:CD4+ T cell restimulation assay over a time span of several years. Therefore, various controls were employed to ensure a consistent and comprehensive analysis of the data.

For each screen, 30 healthy donors were selected based on their HLA-DRB1 alleles to reflect the world population [5] (Appendix A). In addition, a characterization of the dendritic cells (DCs) was included in every screen to assess the phenotype of these cells before and after maturation by flow cytometry. Activation of the DCs was determined by upregulation of key maturation markers on the cell surface that are known to be correlated with T-cell priming capacity: CD40, CD80, CD83, CD86, and HLA-DR [10]. Moreover, CD209, a pathogen-recognition receptor expressed on the surface of immature DCs, is internalized together with other markers, thus resulting in efficient presentation [11]. Accordingly, the downregulation of CD209 is the consequence of a shift from an immature to a mature DC phenotype. A representative distribution of cell surface marker expression at both the immature (iDC) and mature stage (mDC) is shown in Figure 1b. The addition of the DC maturation cocktail, composed of TNF-α and IL-1β, led to a slightly higher expression of CD40, CD80, CD83, and HLA-DR, but also a substantial increase in CD86 expression, resulting in a more than ten-fold increase in the average MFI for this surface marker. In addition, we also observed a moderate decrease in CD209 expression. Altogether, this analysis confirmed that DCs from all donors of the cohort have the potential to be activated prior to their interaction with autologous CD4+ T cells. Moreover, the assay is qualified for a given immunomodulatory protein by treating the DCs together with KLH to assess what impact the protein has on the KLH-induced T cell response. This enables us to highlight proteins that may influence the DC-induced activation of T cells.

### 3.2. DC:CD4+ T Cell Restimulation Assay Precision Assessment and Comparators

We investigated first the repeatability of the assay by testing the IFN-γ response of donors to KLH and Avastin (same production batch) in multiple assay screens. To this aim, we plotted the SI for KLH and Avastin for all donors, grouped by batches. All the donors analyzed over 24 screens consistently showed high SIs (with a geometric mean of 225 across all screens) upon treatment with KLH (a widely accepted positive control), while SIs obtained with treatment with bevacizumab were distributed around 1 (Figure 2a), suggesting that there was no substantial change in IFN-γ release compared to the blank. Moreover, very few donors (40/607, 6.6%) in this treatment group showed a two-fold SI change or above (our criterion for calling a positive response, see Section 2). Based on these findings, we recommend the use of bevacizumab as a negative comparator in this assay. We used KLH as the technical positive control in our analyses, as highly immunogenic biopharmaceuticals tend not to reach marketing authorization [12].

We used the DC:CD4+ T-cell restimulation assay to investigate 24 biotherapeutics developed by a range of pharmaceutical companies, comprising a broad range of drug formats and targets. Details about the molecules were extracted from the corresponding FDA label [13] and are summarized in Table 1.

However, for most of the labels, important data about the trial were missing, ultimately limiting the interpretability of the reported ADA rates. Moreover, for a number of trials, drugs were administered in combination with radiotherapy, which is known to impact the immune system and the subsequent production of ADA [14]. In other cases, biopharmaceuticals were administered with corticoid pre-treatment to dampen the immune response, which also influences the production of ADA. In this manuscript, data from combination trials were omitted, except for Alemtuzumab, Cetuximab, Daratumumab, Elotuzumab, Sarilumab, and Tocilizumab, which are always co-administered with other drugs.

Results are summarized in Figure 2b,c. Most of the tested biopharmaceuticals elicited low levels of IFN-γ release (alirocumab, avelumab, benralizumab, bevacizumab, brentuximab, certolizumab, cetuximab, durvalumab, evolocumab, galcanezumab, necitumumab, nivolumab, sarilumab, tocilizumab, ustekinumab, vedolizumab). However, we saw stronger T cell responses with alemtuzumab, elotuzumab, pembrolizumab, infliximab, and daratumumab, for which more than 10% of the donors showed a SI statistically significant above 2. Interestingly, antibodies with identical modes of action (i.e., infliximab, adalimumab, and certolizumab all target TNF-α) triggered different T cell responses with regards to IFN-γ production. In addition, when compounds were tested several times in different screens, we observed that SIs and the derived response rates showed a significant variability (Figure 2c). We observed that for adalimumab, for which screens 02 and 06 resulted in 23.3% and 26.7% of positive donors, respectively, whereas it dropped to 0% in screen 07. These discrepancies are seen for pembrolizumab, atezolizumab, and elotuzumab, as well. One explanation for this observation could be a compound batch effect, as illustrated for adalimumab in Figure 2d.

### 3.3. Statistical Characterization of the Assay

We investigated the stability of the assay across independent screens and the potential influence of confounding experimental factors using a variance component analysis on the full data set, which included the controls as well as the marketed compounds.

The data reported in this study were recorded in 24 different screens over several years. Hence, data replication occurred on various levels (i.e., bevacizumab and KLH were measured in all screens, several compounds were repeatedly measured in some screens, while subsets of compounds were tested on all donors within each screen), we could estimate the variance contributions of the treatments, the donor, the treatment-donor interaction, and the screen. As a typical screen is done in 3–4 batches, running a given subset of donors on all compounds in each batch, we can also assess the batch effect that is nested in the screen. During the course of the study, healthy donors that had given their blood could visit the blood donation center again, and the derived cells were used in two (or more) screens (i.e., same donor, same treatment but different screens). The results of this analysis are summarized in Figure 3a. Treatment-related effects (the expected effect from a compound in the assay, here driven primarily by the large number of strong KLH responses) accounted for 54% of the total variance; in contrast, the contribution of purely experimental factors was quite small (screen-to-screen variability: 0.5%, batch-to-batch variability within a screen: 2.0%). The donor factor (i.e., a factor accounting for a generally higher or lower donor-specific IFN-γ release independently of the treatment) accounted for 6.9% of the total variability; a similar proportion of the variance (5.4%) was attributed to the donor-treatment interaction (i.e., a factor taking account of a subject-specific response to a given treatment). A relatively high proportion of the total variance (23.6%) could not be readily accounted for by the known experimental factors. This could be due, for example, to the unavoidable technical variability in the protocol used to carry out the assay, or to heterogeneities unaccounted for when collecting sample material from a given donor at different times. In general, it would be very difficult to single out these technical and biological sources of variance and to investigate their relative impact on the assay reproducibility without some very cumbersome additional quality control processes.

The breakdown of the SI readouts into individual variance components enables us to simulate data sets with specified effect sizes for hypothetical treatments. Hence, we can estimate the statistical power (i.e., the probability of detecting a true compound effect) in a wide range of conditions. For example, Figure 3b shows the resulting statistical power when comparing a compound SI fold-change response with the one of a reference or comparator treatment; here, we differentiate the case where both compounds of interest were assessed in the same screen, in contrast to a comparison that was conducted across different screens. A major advantage of a within-screen comparison is that one could apply paired testing (i.e., using ‘donor’ as a covariate) to yield higher statistical power because the donor-to-donor variability would be partially accounted for in this approach. This is, in our opinion, the recommended setting for a compound ranking study. Moreover, depending on the hypothesis of interest, some additional statistical power may be gained by using a one-sided testing approach. This is legitimate when only a higher (or lower) compound response is of interest as compared to a reference treatment, which, in fact, could be the most relevant scenario. As a rule of thumb, we expect that SI differences of about 75% on a linear scale (i.e., a SI fold-change of 1.75 or 0.8 log2 units) can be detected with a statistical power of 80%, assuming one-sided testing within the same screen, alpha = 0.05, and n = 30 donors.

Statistical power is also a function of the sample size (here, number of donors per screen); we next examined this dependency and the impact of this variable in the interpretation of our assay results (Figure 3c). We observe a considerable gain in statistical power for studies including up to 30 donors per study. Increasing the number of donors beyond this point leads to noticeably smaller gains in statistical power at the cost of a considerable increase in effort and expenses, which is associated with larger experiments. In our experience, a standard study size of 30 donors per screen strikes the right balance, both for the experimental and statistical angles. In the case that enhanced statistical power is desired, we believe that a reduction of the residual assay variance by experimental protocol refinements could be a more promising approach than merely increasing the donor count.

### 3.4. Qualification of the Assay Threshold

An essential aspect of the study was to investigate the assay’s ability to predict the potential for unwanted immune responses in line with FDA labels [13]. Accordingly, we characterized our assay in terms of accuracy (overall rate of correct predictions on compound level), sensitivity (probability to detect an immunogenic treatment), specificity (probability of correctly identifying a non-immunogenic treatment), and Positive/Negative Predictive Value (confidence in assigning either label correctly). To this end, we tested the aforementioned 24 molecules for which clinical data were available; however, since ADA responses in a limited number of patients would not necessarily be considered a relevant risk, we divided the tested molecules into two categories: high risk (≥20% reported ADA rate) and low risk (<20% reported ADA rate) for immunogenicity according to the reported data upon treatment. This classification was correlated to the proportion of donors for which a given biopharmaceutical triggered a CD4+ T cell-driven IFN-γ production in the assay: a positive assay readout was set to generate a SI statistically significant above 2 compared to the blank control, while a negative assay readout would not.

Using 10% as an optimal threshold (>3/30 positive donors according to our criteria), the assay reported 4 true positives (TP) and 16 true negatives (TN) for a total of 24 tested biopharmaceuticals (6 categorized as high risk, 18 labeled as low risk). It categorized 2 antibodies (daratumumab and pembrolizumab) at high risk of immunogenicity, even though their clinical ADA rates were below 20% (false positives, FP), while brentuximab and atezolizumab are categorized as low risk of immunogenicity, even though their clinical ADA rate were above 20% (false negatives, FN). The accuracy is the sum of true positives and true negatives over the total of tested compounds, yielding an estimated assay accuracy of 83% (20/24). The sensitivity, TP/(TP + FN), and specificity, TN/(TN + FP), are two additional important estimators, which represent the two types of possible errors. At this threshold, the DC:CD4+ T cell restimulation assay provides a 67% sensitivity at 89% specificity, with a 67% (4/6) and 89% (16/18) Positive and Negative Predictive Value, respectively.

### 3.5. Case Studies in Pre-Clinical Research

An important motivation of running a DC:CD4+ T cell restimulation assay in a pre-clinical setting is to derive information on whether compounds in development might be at risk of inducing an immunogenic response in treated patients. In this context, it is important to reduce false positive compound categorization, even at the expense of a higher false negative rate (i.e., over-classifying new molecules in the high immunogenicity risk category). As part of an integrated immunogenicity risk assessment, other risk factors (e.g., peptide presentation, mode of action, etc.) should also be taken into consideration. Our analysis demonstrates that a direct comparison of the responder rates in the DC:CD4+ T cell restimulation assay with the proportion of ADA-positive patients for a given treatment may not provide the best context of use for this assay. Our proposed strategy is to apply a given threshold to interpret results, essentially reducing the assay output to a binary outcome for biotherapeutics immunogenicity hazard identification. This enables us to retain the essential information on compound risk categorization, while minimizing the impact of noise in the data. Our data suggest that a selected threshold of 10% positive responders to classify a molecule as bearing a higher potential for immunogenicity is the optimal cutoff to flag compounds with high immunogenic potentials, while limiting the number of false negatives at an early stage of preclinical development.

To illustrate the strategy delineated above, we provide here a case study derived from one of our internal programs where seven potential clinical candidates from the same project, which differ from their primary sequence, have been tested in the assay (Figure 4a). The results showed that compounds A, B, D, and G were above the threshold, whereas variants C, E, and F were below the threshold and, therefore, associated with a lower risk of immunogenicity.

Furthermore, we demonstrated that this assay was also suitable for testing whether peptides could trigger a CD4+ T cell response. Hence, we tested known T cell epitopes from Natalizumab and Interferon β, as well as potential deimmunized versions [15,16] (Figure 4b). Peptides were tested at 2 ug/mL and followed the same experimental procedure as described in the Material and Methods section. Results from the assay demonstrate that minor changes in the amino acid sequence of the T cell epitopes could reduce the onset of a CD4+ T cell response, thus confirming the published findings, but also that this assay can accommodate peptides (e.g., peptide based biotherapeutics or T cell epitopes).

## 4. Discussion

The multifactorial nature of immunogenicity requires that an integrated preclinical risk assessment should be a key element of biotherapeutics development. As T cell-dependent responses are major drivers of immunogenicity, in vitro T cell assays are frequently used as tools to identify and measure CD4+ T cell-dependent responses to biotherapeutics. The DC:CD4+ T cell restimulation assay described here assesses the propensity of a biotherapeutic to trigger a CD4+ T cell response that may result in B cell activation and ADA production. This assay plays a key role in our integrated approach to therapeutic protein immunogenicity risk estimation, which could accelerate drug development.

While a number of assays probing T cell activation in the context of immunogenicity have been published in recent years [6], we believe that the DC:CD4+ T cell restimulation assay described here provides a more comprehensive insight into the role of dendritic cells (taken here as the archetypal APC) in the context of their activation of T cells [17]. The immune response follows a three-signal rule for activation (TCR:MHC/peptide interactions, costimulatory interactions such as via CD28, and cytokine production); the assay published herein captures the interplay of all three signals [18]. In addition, the number of preexisting T cells specific to biotherapeutics is very low, ranging between 1 out of 10^8^ (e.g., trastuzumab and etanercept) and 1 out of 10^7^ T cells (e.g., rituximab) [9], but the assay format of the DC:CD4+ T cell restimulation assay allows screening of more CD4+ T cells than in a classical PBMC-based assay. We also believe that the re-stimulation step increases the likelihood of capturing a sustained T cell response [19].

A key part of validating the DC:CD4+ T cell restimulation assay was assessing repeatability and reliability, and its potential to categorize biotherapeutics according to their risk of inducing an unwanted immune response in the clinic. To this end, we carried out a detailed analysis of 24 biotherapeutics with various levels of clinically-detected ADA rates as a proxy for immunogenicity risk. Our robust assessment comprised several levels of repetitions, including repeated assessment of some compounds in multiple assay screens, to provide insights into both the potential and inevitable limitations of the assay. Variance component analysis showed that the primary factors governing the experimental setup, i.e., the screen and the batched donor processing within each screen, did not have a major systematic impact on the readouts. Notable variability arose, however, when compounds were re-tested in another screen, presumably related to subtle variation in compound preparation or the used production batches. Nonetheless, the compound batch effect is not specific to this assay. It is likely that the handling and storage of the sample plays a role here, influencing post-translational modifications and aggregation. Additionally, non-product related factors (e.g., DNA and host cell protein contaminations) have an impact on the risk of immunogenicity and could also influence the assay readout. General donor specific inducibility and the donor specific response to individual compounds also explained parts of the signal variability. However, there was a rather substantial residual unexplained variance, which should caution the user with regard to overinterpretation of individual readouts. Notably, quantification of SI changes in a strict sense were not directly informative, as even clinically-tested compounds with low immunogenic risks (for example, bevacizumab) resulted in a few positive readouts. We presume that additional insights might be gained by fundamental and costly changes of the lab protocol, i.e., performing replicated measurements in different experimental batches for every condition. In our experience, donor cohorts of 30 individuals per screen offer a reasonable tradeoff between the cost, timelines, and statistical power of the assay. While not specifically discussed in the manuscript, we found it important to test all compounds using a panel of donors that showed an HLA-DRB1 allele frequency that broadly reflects the world population. It has been demonstrated that certain HLA alleles were associated with an increased immunogenic response towards certain biopharmaceuticals [20,21,22,23,24,25]. Nevertheless, in the context of use of this preclinical assay carried out in 30 donors, we primarily investigated whether compounds may be at risk of inducing an enhanced immunogenic response in a general population. As each screen usually comprised different sets of donors, an arbitrary selection of pre-typed donors with respect to their allelic HLA-DRB1 composition enabled a higher comparability of the data in the long term.

While tempting, it is problematic to compare SI values measured in a DC:CD4+ T cell restimulation assay with actual ADA rates in the clinic, although it is one of the few available measures directly related to clinical immunogenicity. Assays used to measure ADA in clinics are based on different methodologies sensitive to sample handling, the timing of sample collection, concomitant medications utilized in the study, and the underlying nature of the treated disease [26]. Furthermore, while we believe we have used the most recent information available on the FDA database, most labels may not be updated on a regular basis: in a recent review, Borrega et al. showed that 57% (39/69) of the biological drugs authorized before 2012 did not have updated summaries of product characteristics, especially in the immunogenicity section [27]. In our study, we collected the ADA rates of the 24 assessed marketed compounds as a starting point to build a database to benchmark newly developed immunogenicity estimation methods and to have a retrospective and comprehensive overview of the immunogenicity of marketed antibodies. We used the available data to create two categories of compounds, at high (≥20% reported ADA rate) and low (<20% ADA rate) risk for immunogenicity, on which we calibrated the assay’s linear mixed model. Thus, this binary high/low risk paradigm is the most reasonable for implementation in preclinical risk evaluation for therapeutics. To facilitate this process, we found it essential to add in the panel a few standard compounds (at minimum, a negative control, such as bevacizumab, and a positive control, such as KLH; any additional comparators also provide useful comparisons) to help set precise boundaries of low and high risk of immunogenicity while mitigating intrinsic donor variability. Accordingly, in this context of use, one of the most useful applications of the DC: CD4+ T cell restimulation assay is to provide a relative ranking for compounds with similar amino acid sequences and mode of action, or compounds that have different formulation or have been produced in different batches.

While assays measuring T cell activation in response to novel biopharmaceuticals are not yet required by regulatory agencies, there is added value in presenting the results of such assays as part of the risk assessment submitted in the Integrated Summary of Immunogenicity [28]. A current challenge is that none of the published assays is considered to be fully validated. We here propose a new assay format that captures the interaction between DCs and CD4+ T cells by monitoring the production of IFN-γ by CD4+ T cells in response to biotherapeutics processed by DCs. We tested the predictive power of this assay vs. clinical ADA rate by assessing 24 marketed antibodies, which resulted in 83% accuracy. Predicting the actual rate of ADA-positive patients in a clinical setting with a single in vitro assay is unlikely to be possible, given the myriad contributing factors. However, the DC:CD4+ T cell restimulation assay can help flag potentially immunogenic biopharmaceuticals in preclinical drug development, allowing for selection or de-immunization before a clinical trial starts, improving both patient safety and the cost of pharmaceuticals. Implementation of this assay as part of a comprehensive risk assessment has the potential to provide a more robust and informative immunogenicity risk assessment in early drug development.

## Figures and Tables

**Figure 1 pharmaceutics-14-02672-f001:**
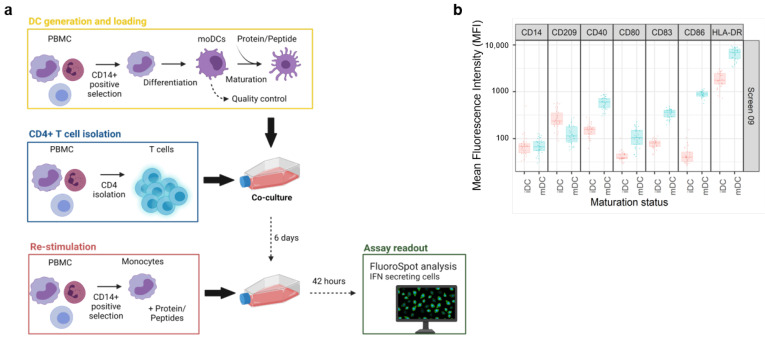
The DC:CD4+ T cell restimulation assay. (**a**) Experimental setup of the DC:CD4+ re-stimulation assay. The assay starts with the isolation of monocytes from healthy donor PBMCs, followed by the loading of the protein of interest and maturation of the monocyte derived Dendritic Cells (moDC). Autologous CD4+ T cells are isolated and co-cultured with the loaded moDCs. After an incubation of 6 days, freshly isolated monocytes are challenged with the same protein and added to the co-culture for an additional 42 h before analyzing the production of IFN-γ by FluoroSpot. (**b**) DC were characterized by the expression of the following cell surface markers: CD11c, CD14, CD80, CD83, CD86, CD209, and HLA-DR before and after DC activation by addition of TNF-α and IL-1β to ensure good cell fitness. Created with BioRender.com, accessed on 27 November 2022.

**Figure 2 pharmaceutics-14-02672-f002:**
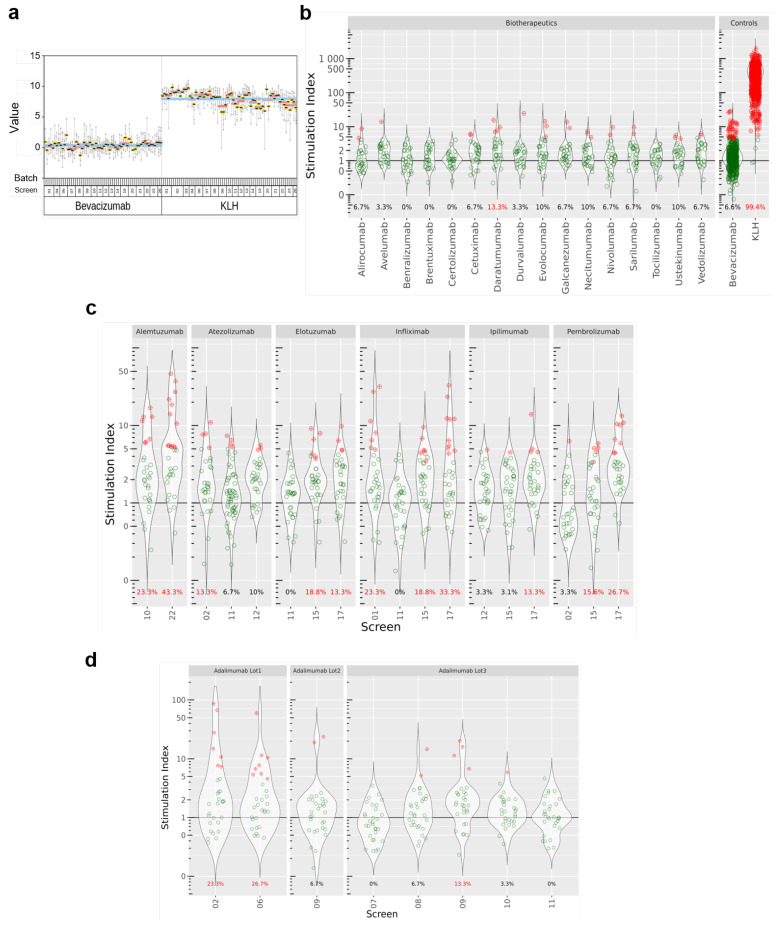
Overview of the stimulation indexes (SI) obtained in the DC:CD4+ T cell restimulation assay. (**a**) Stability of the controls over different assay screens and donor batches. Data were generated for the set of benchmark compounds in single (**b**) or multiple screens (**c**,**d**). SI represents the number of IFN-γ positive cells over baseline. If a datapoint is significantly above the SI threshold of 2, the donor is considered as positive for this condition and appears in red.

**Figure 3 pharmaceutics-14-02672-f003:**
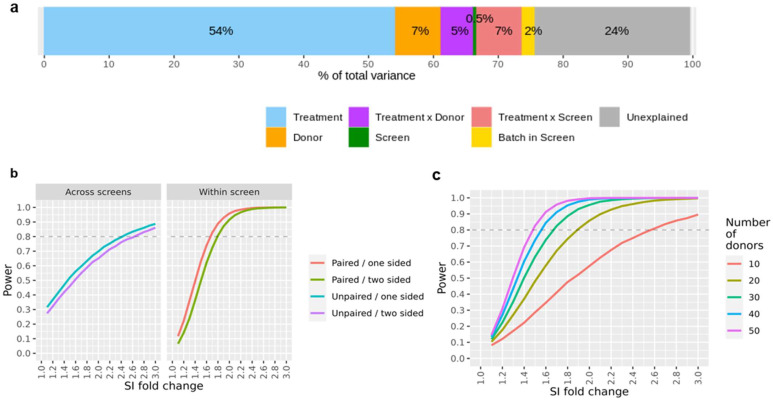
Variance and assay power. (**a**) Main factors contributing to assay variability estimated by a variance component analysis. The fitted model is the following: log2 (SI) ~ (Compound × DonorID) + Screen/Batch. There is a relatively low relative impact of assay batching variables (Screen, Batch within a screen) in comparison to the compound component. (**b**) DC:CD4+ T cell restimulation assay power curves for compound comparison. (**c**) Assay power curves showing the statistical power to detect a treatment effect by comparing a compound with a comparator treatment. A one-sided paired test within-study (α = 0.05) according to various donor cohort sizes has been used.

**Figure 4 pharmaceutics-14-02672-f004:**
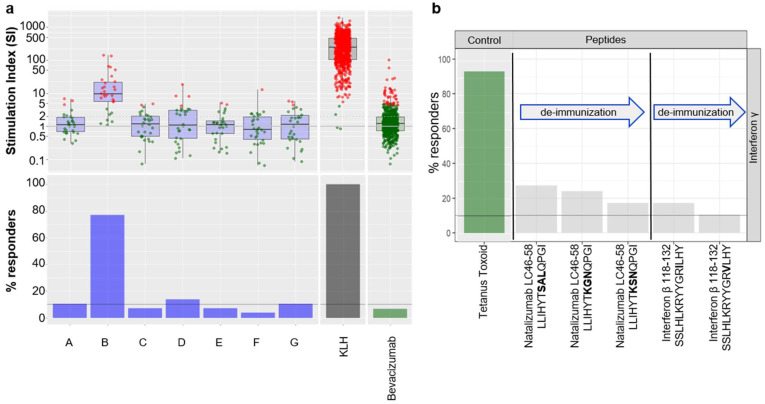
Case studies in pre-clinical research. (**a**) Stimulation Index (SI) obtained for 7 different candidate compounds of the same project (named A to G). The change in color, from green to red, depicts the positivity of the donor within the screen. The lower panel represents the proportion of positive donors. The threshold derived from the validation study is set at 10% positive donors. (**b**) DC:CD4+ re-stimulation results obtained for a selection of known T cell epitopes derived from biotherapeutics and their “de-immunized” counterparts.

**Table 1 pharmaceutics-14-02672-t001:** Overview of the test items and their respective clinical ADA rates. Alemtuzumab, cetuximab, daratumumab, elotuzumab, sarilumab and tocilizumab are part of a co-treatment. Therefore, consideration should be taken when looking at the reported ADA rates. The information was extracted from FDA labels [13]. If several clinical ADA rates were reported, studies mentioning a co-treatment were excluded and the mean value for the remaining study outcomes was taken. In many cases, larger deviations may be due to systematic differences in the treated patient populations, as well as different analytical methods.

Antibody Name	Trade Name	Format	Target	Main Target Patient Population	Clinical ADA Rate	Screens
Adalimumab	Humira	Human IgG1	TNF-α	Rheumatoid Arthritis	23	02; 06; 07; 08; 09; 10; 11
Alemtuzumab	Lemtrada	Humanized IgG1	CD-52	Multiple Sclerosis	35	10; 22
Alirocumab	Praluent	Human IgG1	PCSK9	Cardiovascular disease	5	10
Atezolizumab	Tecentriq	Human IgG1 no-Glyco	PD-L1	Non-Small-Cell Lung Carcinoma (NSCLC)	44	02; 11; 12
Avelumab	Bavencio	Human IgG1	PD-L1	Urothelial Carcinoma	17	12
Benralizumab	Fasenra	Humanized IgG1	CD-125	Asthma	13	11
Bevacizumab	Avastin	Humanized IgG1	VEGF	Solid Tumor	0.6	ALL
Brentuximab	Adcetris	Chimeric IgG1-ADC	CD-30	Classical Hodgkin Lymphoma (late stage)	30	11
Certolizumab	Cimzia	FabPEG	TNF-α	Crohn Disease and Rheumatoid Arthritis	8	10
Cetuximab	Erbitux	Chimeric IgG1	EGFR	Head, Colorectal and Neck Cancer	5	12
Daratumumab	Darzalex	Human IgG1	CD-38	Multiple myeloma	0	12
Durvalumab	Imfinzi	Human IgG1	PD-L1	Locally advanced or Metastatic Urothelial Carcinoma, NSCLC	3	10
Elotuzumab	Empliciti	Human IgG1	SLAMF7	Multiple Myeloma	27	11; 15; 17
Evolocumab	Repatha	Human IgG2	PCSK9	Cardiovascular Disease	0.3	10
Galcanezumab	Emgality	Humanized IgG4	Calcitonin	Migraine	5	10
Infliximab	Remicade	Chimeric IgG1	TNF-α	Psoriatic Arthritis	27	01; 11; 15; 17
Ipilimumab	Yervoy	Human IgG1	CTLA-4	Metastatic melanoma, advanced renal cell carcinoma, metastatic colorectal cancer	8	12; 15; 17
Necitumumab	Portrazza	Human IgG1	EGFR	NSCLC	4	12
Nivolumab	Opdivo	Human IgG4-CPPC	PD-1	NSCLC	11	02
Pembrolizumab	Keytruda	Humanized IgG4-CPPC	PD-1	Cancer	2	02; 15; 17
Sarilumab	Kevzara	IgG1	IL-6R	Rheumatoid Arthritis	9	10
Tocilizumab	Actemra	Humanized IgG1	IL-6R	Rheumatoid Arthritis	2	10
Ustekinumab	Stelara	Human IgG1	IL-12/IL-23	Plaque Psoriasis	6	10
Vedolizumab	Entyvio	Humanized IgG1	Integrin α4β7	Ulcerative colitis and Crohn’s disease	6	11

## Data Availability

Data are contained within the article or Appendix A.

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
