# Peer review of "Validation of a Dendritic Cell and CD4+ T Cell Restimulation Assay Contributing to the Immunogenicity Risk Evaluation of Biotherapeutics"

_pharmaceutics, 2022, doi:10.3390/pharmaceutics14122672_

Round 1

Reviewer 1 Report

The manuscript entitled “Validation of a dendritic cell and CD4+ T cell restimulation assay contributing to the immunogenicity risk evaluation of biotherapeutics” describe the set up and validation of a FluoroSpot-based in vitro assay for the evaluation of drug-specific T cell responses considering a panel of 24 biotherapeutics. The authors demonstrate that the dendritic cell and CD4+ T cell restimulation assay is a valuable tool to assess the risk of drug-specific T cell responses.

The manuscript is fluent and the subject is original and of interest to readers.

The work can be improved in some points. Below are my suggestions to the authors:

- I have some questions about the in vitro use of biotherapics: how did the authors optimize the concentration of biotherapics for the DC load? Was it optimized for each individual compound or did they always use the same concentration? Is it possible that the "negativity" of some test is due to concentration problems? Furthermore: have the biotherapeutics been used pure or in their pharmaceutical form (with excipients)? Authors should specify in the "methods" section.

- Another question about biotherapeutics: mabs used for the incubation of APCs (DCs or monocytes) can have a direct impact on the functionality of the APCs themselves. For example, could Ipilimumab (anti-ctla4) induce the production of anti-inflammatory cytokines by DCs with a negative impact on the final result of the re-stimulation test? Authors should discuss this hypothesis if relevant.

- Some details are missing in the restimulation assay method: the CD4/DC ratio at day 0; the Monocytes/CD4+DC ratio on day 6; the cell culture media used (with or without serum?). Do the authors only use cryopreserved PBMCs, or also fresh ones? Authors should specify in the "methods" section.

- Correctly, the authors were very detailed in describing the analysis setup, the statistics used and the validation parameters they considered; however, no mention is made of the spot quantification method. Is the quantification automated? This description may be important especially in light of what the authors have discussed regarding "However, there was a rather substantial inexplicable residual variance, which should warn the user about overinterpreting individual reads". Authors should describe and specify in the "methods" section.

- One more question about the data acquisition method: why did the authors choose to use the fluorospot technique instead of a more widespread technique such as, for example, flow cytometry? Authors should discuss this briefly, at least in the introduction. 

- What do the authors mean by "Test items were studied in independent DC screens: CD4+ T cell restimulation assay over a period of several years"?

Author Response

The manuscript entitled “Validation of a dendritic cell and CD4+ T cell restimulation assay contributing to the immunogenicity risk evaluation of biotherapeutics” describe the set up and validation of a FluoroSpot-based in vitro assay for the evaluation of drug-specific T cell responses considering a panel of 24 biotherapeutics. The authors demonstrate that the dendritic cell and CD4+ T cell restimulation assay is a valuable tool to assess the risk of drug-specific T cell responses.

The manuscript is fluent and the subject is original and of interest to readers.

The work can be improved in some points. Below are my suggestions to the authors:

- I have some questions about the in vitro use of biotherapics: how did the authors optimize the concentration of biotherapics for the DC load? Was it optimized for each individual compound or did they always use the same concentration? Is it possible that the "negativity" of some test is due to concentration problems? Furthermore: have the biotherapeutics been used pure or in their pharmaceutical form (with excipients)? Authors should specify in the "methods" section.

The aim is to try and expose the DC to as much product as possible to maximize the amount of protein being taken-up, processed and presented by the DC. In this regard, negative results due to concentration are unlikely. However, we also have to balance the fact that many proteins being tested at the early stages of development may only be available at relatively low concentrations (sometimes < 1 mg/mL), which limits the maximum concentration that can be added to the DC. We have carried out a number of experiments to assess the minimum concentration of protein required to detect a response in the assay and this is in the region of 50 µg/mL (300 nM) for a typical monoclonal antibody. Increasing the concentration above this level does not typically lead to an increase in response, however therapeutic antibody concentrations lower than 10 µg/mL (60 nM) can sometimes lead to lower responses. The majority of biotherapeutics used to benchmark the assay are the exact product batches that are administered to patients in the clinic (“bought from Runge Pharma GmbH & Co in their respective formulation”). This information was added to the manuscript.

- Another question about biotherapeutics: mabs used for the incubation of APCs (DCs or monocytes) can have a direct impact on the functionality of the APCs themselves. For example, could Ipilimumab (anti-ctla4) induce the production of anti-inflammatory cytokines by DCs with a negative impact on the final result of the re-stimulation test? Authors should discuss this hypothesis if relevant.

The assay is often qualified for each individual biotherapeutic being tested to ensure that there is no interference from the protein mode of action. It is true that some immunomodulatory proteins can give rise to false-positives or false-negatives depending on what receptors are targeted on the surface of the DCs. During the assay qualification the DCs are treated with the immunomodulatory protein and a positive control (e.g. KLH) to assess what impact the protein has on the KLH-induced T cell response. This enables us to highlight proteins that may inhibit the DC-induced activation of T cells and also those proteins which activate the DC and/or T cells independently of HLA:peptide:TCR signaling. There are also additional assays that can be used to complement this assessment to ensure that in the DC:CD4 re-stimulation assay we are only detecting HLA:peptide-induced T cell activation (e.g. DC activation assays to assess DC cell surface markers by FACS and/or cytokine release by Luminex®). This point has been clarified in the manuscript (3.1. DC:CD4+ T cell restimulation assay workflow).

- Some details are missing in the restimulation assay method: the CD4/DC ratio at day 0; the Monocytes/CD4+DC ratio on day 6; the cell culture media used (with or without serum?). Do the authors only use cryopreserved PBMCs, or also fresh ones? Authors should specify in the "methods" section.

The DC:CD4 ratio is 1:10 and the CD14:CD4 ratio is also 1:10. The assay medium used is serum-free (Cellgenix DC medium from Cellgenix). All PBMC used in these assays are isolated from fresh whole blood or leukopaks and cryopreserved within 4-6 hours of collection from the donor. PBMC are controlled-rate frozen and stored in vapor-phase nitrogen at -196°C. These details have been added in the manuscript (see 2. Materials and Methods)

- Correctly, the authors were very detailed in describing the analysis setup, the statistics used and the validation parameters they considered; however, no mention is made of the spot quantification method. Is the quantification automated? This description may be important especially in light of what the authors have discussed regarding "However, there was a rather substantial inexplicable residual variance, which should warn the user about overinterpreting individual reads". Authors should describe and specify in the "methods" section.

The SFU (spot forming units) in each well of the FluoroSpot plate were calculated using the IRIS FluoroSpot reader (Mabtech). This high resolution reader and associated software allows the automation of the spot counting and reduces user bias in determining ‘real’ spots from the background. These details have been added in the manuscript (see 2.3. DC:CD4+ re-stimulation assay (Epibase®IV, Lonza))

- One more question about the data acquisition method: why did the authors choose to use the fluorospot technique instead of a more widespread technique such as, for example, flow cytometry? Authors should discuss this briefly, at least in the introduction. 

When developing the DC:CD4 re-stimulation assay a variety of readout methods were evaluated. In particular, FluoroSpot was directly compared to T cell proliferation by FACS. It was shown that FluoroSpot tended to be the most more sensitive method, at least in part because this technology allows us to assess all of the T cells in each well (rather than only or portion of the cells by FACS) and also because cytokine release is a more immediate response than subsequent proliferation.

- What do the authors mean by "Test items were studied in independent DC screens: CD4+ T cell restimulation assay over a period of several years"?

 “independent screens of the DC:CD4+ T cell restimulation assay over a time span of several years.”. This sentence highlights the duration of the study and the fact that the 24 tested biotherapeutics were distributed in different screens. Therefore, each study used PBMC from different donors and was often focused on different sets of test proteins. 

Reviewer 2 Report

In this manuscript, Siegel et al described a functional assay to evaluate the immunogenicity risk of biotherapeutics. The manuscript is very interesting, it describes a technology that meets a need for which there is no solution to date and is overall very pleasant to read.

I have specific points to address:

- Three monoclonal antibodies were produced in-house (briakinumab, secukinumab and ixekizumab). It is therefore possible that the immunogenicity of these mAbs is not identical to the commercialized mAbs. It therefore seems important to mention it in the manuscript

- Subsection 2.2. Healthy donor cohort of the Materails & Methods: Specify the approval number of the local Research Ethics Committee

- Subsection 2.3. DC:CD4+ re-stimulation assay (Epibase®IV, Lonza). Indicate the concentration of protein/peptide used to load iDCs.

- Subsection 2.3. DC:CD4+ re-stimulation assay (Epibase®IV, Lonza). “The mDCs were then co-cultured with 1.2 million autologous CD4+ T cells”.  Specify the coculture conditions (how many DCs versus how many T cells in which type of plate and indicate the final volume). Also indicate the exact composition of the culture medium. At day 5, is there washing of the DCs before coculture with the T cells, or are the T cells added directly to the DCs in the presence of GM-CSF and IL-4?

- In the same subsection: As I understand, the authors use monocytes loaded with peptide/proteins to restimulate cocultures? The loaded monocytes (how many, at what concentration?) are placed in Fluorospot plates after 4 hours of incubation and the DC/T cell cocultures are then added to these plates? This methodological aspect should be clarified.

. In my opinion, it cannot be excluded in this functional assay that the Mo-DCs differentiated in the presence of GM-CSF and IL-4 present peptides of these two cytokines and stimulate autoreactive CD4+ T cells specific for these 2 cytokines. Indeed, it is not uncommon to find autoantibodies directed against these 2 cytokines in many autoimmune contexts. What negative control is used? unloaded mature DCs versus loaded mature DCs or unloaded immature DCs versus loaded mature DCs?

- Figure 1: MFI of MHC class II expression on iDCs seem to be huge. Please indicate in the Mat & Meth section the references of all the Abs used for the phenotype. It could be interesting to show a representative phenotype of both iDCs and mDCs. Incidentally, I congratulate the authors because for my part I have never succeeded in generating satisfactory Mo-DCs from frozen monocytes.

- Figure 2D : The “compound batch effect” seems particularly important. This is an important point that should be detailed in the Discussion section. I did note that the authors allude to it in the Discussion section, but it should be highlighted more and deserves to be more detailed because it constitutes an unexpected major issue of the technique, unless one wishes to compare the immunogenicity of several batches of the same compound. 

Author Response

In this manuscript, Siegel et al described a functional assay to evaluate the immunogenicity risk of biotherapeutics. The manuscript is very interesting, it describes a technology that meets a need for which there is no solution to date and is overall very pleasant to read.

I have specific points to address:

- Three monoclonal antibodies were produced in-house (briakinumab, secukinumab and ixekizumab). It is therefore possible that the immunogenicity of these mAbs is not identical to the commercialized mAbs. It therefore seems important to mention it in the manuscript

We apologize for accidentally citing briakinumab, secukinumab and ixekizumab in the Material and Methods section and would like to highlight that they were not included in the study.

We agree with the reviewer’s comment that non-product related factors (e.g., impurities, post-translational modifications) have an impact on the risk of immunogenicity and could also influence the assay readout. This point was emphasized in the manuscript (see 3.3. Statistical characterization of the assay).  

- Subsection 2.2. Healthy donor cohort of the Materails & Methods: Specify the approval number of the local Research Ethics Committee

The local research ethics committee (REC) reference number is 21/LO/0474. This information has been added in the manuscript (see 2. Materials and Methods)

- Subsection 2.3. DC:CD4+ re-stimulation assay (Epibase®IV, Lonza). Indicate the concentration of protein/peptide used to load iDCs.

Whole proteins were tested at a final concentration of 300 nM (approximately 50 µg/mL for a monoclonal antibody). Peptides were tested at a concentration of 10 µg/mL. This information was added to the “2.1. Compounds” section.

- Subsection 2.3. DC:CD4+ re-stimulation assay (Epibase®IV, Lonza). “The mDCs were then co-cultured with 1.2 million autologous CD4+ T cells”.  Specify the coculture conditions (how many DCs versus how many T cells in which type of plate and indicate the final volume). Also indicate the exact composition of the culture medium. At day 5, is there washing of the DCs before coculture with the T cells, or are the T cells added directly to the DCs in the presence of GM-CSF and IL-4?

1:10 ratio of DC to T cells (100,000 DC and 1,000,000 CD4+ T cells) are co-cultured in deep-well plates (Greiner cat: 780271) in a volume of 1.2 mL assay media (Cellgenix DC medium, serum-free, supplemented with 0.05 mg/mL Gentamicin Lonza Cat: 17-518L). The DCs are washed 5 times after loading with the test protein and then another 5 washes after maturation to remove the cytokines prior to T cell co-culture. These points have been clarified in the Material and Methods section.

- In the same subsection: As I understand, the authors use monocytes loaded with peptide/proteins to restimulate cocultures? The loaded monocytes (how many, at what concentration?) are placed in Fluorospot plates after 4 hours of incubation and the DC/T cell cocultures are then added to these plates? This methodological aspect should be clarified.

During restimulation, monocytes are loaded for 4 hours with the test protein/peptide and then washed 5 times prior to loading onto the FluoroSpot plates with the T cells from the initial co-culture. 25,000 monocytes are added per well along with 250,000 CD4+ T cells from the co-culture (1:10 ratio) in quadruplicate onto the FluoroSpot plate (Mabtech Cat: FSP-0108-10 with final volume of 200 ul/well). These clarifications have been added to the manuscript

In my opinion, it cannot be excluded in this functional assay that the Mo-DCs differentiated in the presence of GM-CSF and IL-4 present peptides of these two cytokines and stimulate autoreactive CD4+ T cells specific for these 2 cytokines. Indeed, it is not uncommon to find autoantibodies directed against these 2 cytokines in many autoimmune contexts. What negative control is used? unloaded mature DCs versus loaded mature DCs or unloaded immature DCs versus loaded mature DCs?

Unloaded (assay medium only) DCs are always included as a control in every donor. These are matured with cytokines in the same way as the conditions treated with the test proteins. In general, very few activated T cells are detected in these unloaded conditions. This data point is then used to calculate the Stimulation Index (SI) and refers to unloaded mature DCs versus loaded mature DCs. The assay has also been validated with other human proteins like albumin and very few responses are detected against fully human proteins.

- Figure 1: MFI of MHC class II expression on iDCs seem to be huge. Please indicate in the Mat & Meth section the references of all the Abs used for the phenotype. It could be interesting to show a representative phenotype of both iDCs and mDCs. Incidentally, I congratulate the authors because for my part I have never succeeded in generating satisfactory Mo-DCs from frozen monocytes.

Thank you for your kind words. The moDC we generate always expresses relatively high levels of HLA-DR which tend to upregulate upon maturation. The anti-HLA-DR-BV605 antibody used in the assay is the L243 clone from Biolegend. Other antibody clones used are as follows: CD11c – 3.9, CD14 – 63D3, CD40 – 5C3, CD80 – 2D10, CD83 – HB15E, CD86 – BU63, CD209 – 9E9A8. A representative phenotype of iDCs and mDCs is illustrated on figure 1.b. The details on the antibody clones have been added to the Material and Methods section.

- Figure 2D : The “compound batch effect” seems particularly important. This is an important point that should be detailed in the Discussion section. I did note that the authors allude to it in the Discussion section, but it should be highlighted more and deserves to be more detailed because it constitutes an unexpected major issue of the technique, unless one wishes to compare the immunogenicity of several batches of the same compound

We agree with the reviewer’s comment and have elaborated on this topic in the discussion.

Round 2

Reviewer 1 Report

Thank you for taking my questions and queries seriously and addressing most of them. I think the manuscript is suitable for publication. Congratulations on an interesting study.